# Genetic diversity and drug resistance profiles of *Mycobacterium tuberculosis* among Ethiopian children as determined by whole-genome sequencing

Yeshiwork Abebaw,[1,2] Abaysew Ayele,[3] Dawit Hailu Alemayehu,[3] Gebremedhin Gebremicael,[2] Getu Diriba,[2] Atsbeha Gebreegziabxier Weldemariam,[2] Betselot Zerihun,[2] Shewki Moga,[2] Bethlehem Adnew,[3] Kalkidan Melaku,[3] Muluwork Getahun,[2] Rahel Argaw,[4] Anandi Sheth,[5] Markos Abebe,[3] Woldaregay Erku Abegaz[1]

**ABSTRACT** Ethiopia ranks 30th among the tuberculosis (TB) burden countries, with children representing a significant yet understudied population group. This study aims to investigate the genetic diversity and drug-resistant profile among Ethiopian children. We included children under 15 years of age diagnosed with culture-confirmed pulmonary TB/drug-resistant TB between January 2017 and June 2023. Phenotypic drug susceptibility testing and whole-genome sequencing were conducted for 85 *Mycobacterium tuberculosis* (MTB) isolates. Demographic data were combined with genomic information. Lineage 4 was the most dominant (77.6%), while lineage 2 was less common (1%). Within lineage 4, several sub-lineages were identified, with lineage 4.2.2.2 being notably the most predominant (48%). Most of these cases were from Oromia (58%), including the hotspot areas for lineage 4 that were identified at a 99% confidence level. Among 17 MDR/pre-XDR-TB isolates, lineages 3 and 4.2.2.2 were the dominantly observed lineages/sub-lineages, with proportions of 29% and 65%, respectively. Of the 85 cases, 30.5% were drug-resistant TB to at least one of the five first-line anti-TB drugs tested by phenotypic drug susceptibility testing. Of these 26 drug-resistant TB cases, 23 were concordant with whole-genome sequencing characterization. The most frequent resistance mutations to rifampicin were found in the *rpoB* gene, specifically p.Ser450Leu (88%), followed by isoniazid in the *katG* gene, p.Ser315Thr (86%). Multidrug-resistant TB was strongly associated with MTB lineages ($P = 0.007$). This study identified high genetic diversity of *M. tuberculosis* and related drug-resistance mutations, with a strong concordance between whole-genome sequencing-based predictions and phenotypic drug susceptibility testing.

**IMPORTANCE** Our findings revealed a high genetic diversity of *Mycobacterium tuberculosis* among Ethiopian children, with the most common lineage being lineage 4, specifically lineage 4.2.2.2, in which a higher frequency of multidrug-resistant tuberculosis (TB) was observed. Additionally, we identified regional hotspots, suggesting ongoing community transmission. Moreover, whole-genome sequencing demonstrated high concordance with phenotypic drug susceptibility testing and identified mutation genes associated with first- and second-line anti-TB drugs, highlighting its usefulness in providing comprehensive results for resistance detection in children. Thus, it is essential for integrating genomic surveillance into childhood TB and drug resistance control.

**KEYWORDS** children, Ethiopia, region, phenotypic drug susceptibility testing, tuberculosis, whole-genome sequencing

**Peer Reviewers** Hamza Babiker, Sultan Qaboos University College of Medicine and Health Science, Muscut, Oman; Hao Li, China Agricultural University, Beijing, China

Address correspondence to Yeshiwork Abebaw, yabebaw18@gmail.com.

The authors declare no conflict of interest.

See the funding table on p. 13.

Drug-resistant tuberculosis (DR-TB) remains a significant public health challenge in the world. In 2024, the World Health Organization (WHO) classified DR-TB as one of the critical priority pathogens posing a severe threat to human health in the fight against antimicrobial resistance (1). Over the past two decades, surveillance of DR-TB has been influential in shaping the response to the DR-TB epidemic (2). Moreover, according to the 2025 WHO report, childhood tuberculosis (TB) accounts for 11% of the global burden of TB cases (3).

Ethiopia is 1 of the 30 countries identified as having a high burden of TB and TB/HIV co-infections (4, 5). However, since 2020, the country has been removed from the list of countries with high rifampicin-resistant (RR) and multidrug-resistant tuberculosis (MDR-TB) burden (4, 5). According to the 2019 drug resistance survey, 1.1% of new TB cases and 7.36% of previously treated cases in Ethiopia were identified as RR/MDR-TB (4, 5). However, the COVID-19 pandemic and the country's political instability could reverse this progress (6). During the COVID-19 pandemic, RR increased by 27.7% compared to the pre-COVID-19 era (6). Additionally, children represent a significant yet understudied group (5). The annual estimation of childhood TB in Ethiopia is high, accounting for 11% of the total estimated TB cases in 2022 (7). A sub-analysis of the third-round drug resistance survey revealed that MDR-TB was 1.3% among Ethiopian children (4).

Drug-resistant tuberculosis in children is typically transmitted through prolonged close contact with an adult who has DR-TB (8). Young children are particularly vulnerable to developing active and disseminated TB due to the immaturity of their immune cells (9). Similarly, children infected with HIV and those born in countries with high TB prevalence are at a higher risk of developing TB within 1 year of infection (10, 11).

*Mycobacterium tuberculosis* (MTB) genotypes are geographically structured into nine major tuberculosis lineages (L) and several sub-lineages (12). Among these, L-1, L-5, and L-6 are ancient lineages, with *Mycobacterium canetti* as the more ancestral branch. At the same time, L-2, L-3, and lineage 4 (L-4) are classified as modern lineages, and L-7 appears to be intermediate between the ancient and modern lineages. L-4, which originated in Europe and the Middle East, is the most widely distributed globally, including in Ethiopia, which has a long history of interactions with these regions (13, 14). On the other hand, L-3 originated in India and Central Asia, areas with which Ethiopia has had a bi-directional socioeconomic relationship, and is potentially spreading from India to Ethiopia (14). However, L-7 remains restricted to Ethiopia (15).

Molecular typing is central to TB molecular epidemiology, with each method offering specific strengths and limitations. Conventional genotyping methods, such as spoligotyping and mycobacterial interspersed repetitive unit (MIRU)-variable number tandem repeat methods, are cost-effective and reproducible; however, they provide limited discriminatory power due to their focus on selected genetic markers (16). In contrast, next-generation sequencing, such as whole-genome sequencing (WGS), offers genome-wide resolution for detailed epidemiological and resistance analyses, although its implementation remains limited in resource-constrained settings (17).

In Ethiopia, many studies have focused on the genetic diversity of MTB using conventional methods (18, 19), yielding limited genome-level data on circulating lineages, drug resistance patterns, and associated mutations in childhood TB. Although genomic surveillance is essential for childhood TB management and drug-resistance control strategies, data specific to childhood TB in Ethiopia remain scarce. Thus, this study aims to investigate the genetic diversity of the MTB lineage and drug resistance prediction based on WGS. We are also identifying drug resistance genes from WGS, taking the phenotypic TB drug sensitivity assay (drug susceptibility testing [DST]) as the gold standard.

## RESULTS

### Sociodemographic characteristics

A total of 105 culture-confirmed childhood TB cases were identified during the study period, with 85 isolates having genomic data included in the genomic analysis. Most of the study participants were in the 10- to 14-year age range (64/85, 75%), with a median age of 12. In this study, the number of girls was slightly higher than that of boys, and most of them were new cases (76, 88.4%) (Table 1).

### Lineage diversity and clustering

This study identified a high level of genetic diversity. The phylogenetic analysis revealed the presence of L-2, L-3, L-4, and L-7. L-4 was the most predominant (66/85, 77.6%), followed by L-3 (16/85, 18.8%). L-7 and L-2 were less common (2, 2.4%, and 1, 1.2%, respectively). Within L-4, several sub-lineages were identified. Sub-lineage 4.2.2.2 accounted for the highest proportion among L-4 (32/66, 48%) and total lineages (32/85, 37.6%). The other identified L-4 sub-lineages were L-4.1.1.3 (2/66, 3%), L-4.1.2.1 (5/66, 7.5%), L-4.1.4 (1/66, 1.5%), L-4.2.1 (10/66, 15.2%), L-4.6.3 (5/66, 7.5%), and L-4.8 (6/66, 9%) (Fig. 1).

Cluster analysis with a predefined cut-off distance of less than 50 single-nucleotide polymorphisms (SNPs) between isolates showed 34 isolates in nine clusters (clustering rate: 40%), distributed across the regions of Ethiopia. Worth noting in this regard is that only two isolates from lineage 4.7 exhibited a genetic distance of six SNPs: sample ET1447 from Addis Ababa, Gulele *wereda* 3, and CH70 from Oromia, West Wollega, Guliso *wereda* (Fig. 2); unfortunately, the epidemiological lineage data of these two cases were not available. In lineage 7, two strains clustered together at a 31 SNP distance. However, the residences of both cases were within 5 km of each other (North Shewa, Oromia, Wuchale *woreda*), which may indicate an epidemiological link that could be considered as evidence for community transmission. The cases involve two males, aged 12 and 10 years. Their samples were also collected in the same month of 2020, and their TB drug resistance profile for both cases showed susceptibility to the first-line drugs tested.

### Geographic distribution of MTB strains

The geographic distribution of the dominant tuberculosis lineages from this study was mainly observed in Oromia (49/85, 57.6%) and the Southern Nations, Nationalities, and Peoples' Region (SNNP) (26/85, 30.5%), as the majority of these cases were collected

**TABLE 1** Sociodemographic characteristics of childhood TB

| Variables | | *n* | % |
|---|---|---|---|
| Age | 0–4 | 8 | 9.5 |
| | 5–9 | 13 | 15.3 |
| | 10–14 | 64 | 75.2 |
| Sex | Male | 41 | 48.2 |
| | Female | 44 | 51.8 |
| History of TB treatment | New | 76 | 88.4 |
| | Had previous treatment | 9 | 11.6 |
| HIV | Positive | 5 | 5.9 |
| | Negative | 80 | 94.1 |
| Contact | Yes | 50 | 58.8 |
| | No | 30 | 35.3 |
| | Unknown | 5 | 5.9 |
| Phenotypic MDR-TB | Resistance | 18 | 21 |
| | Susceptible | 67 | 79 |
| Residence | Urban areas | 25 | 29.4 |
| | Rural areas | 60 | 70.6 |
| Total | | 85 | 100 |

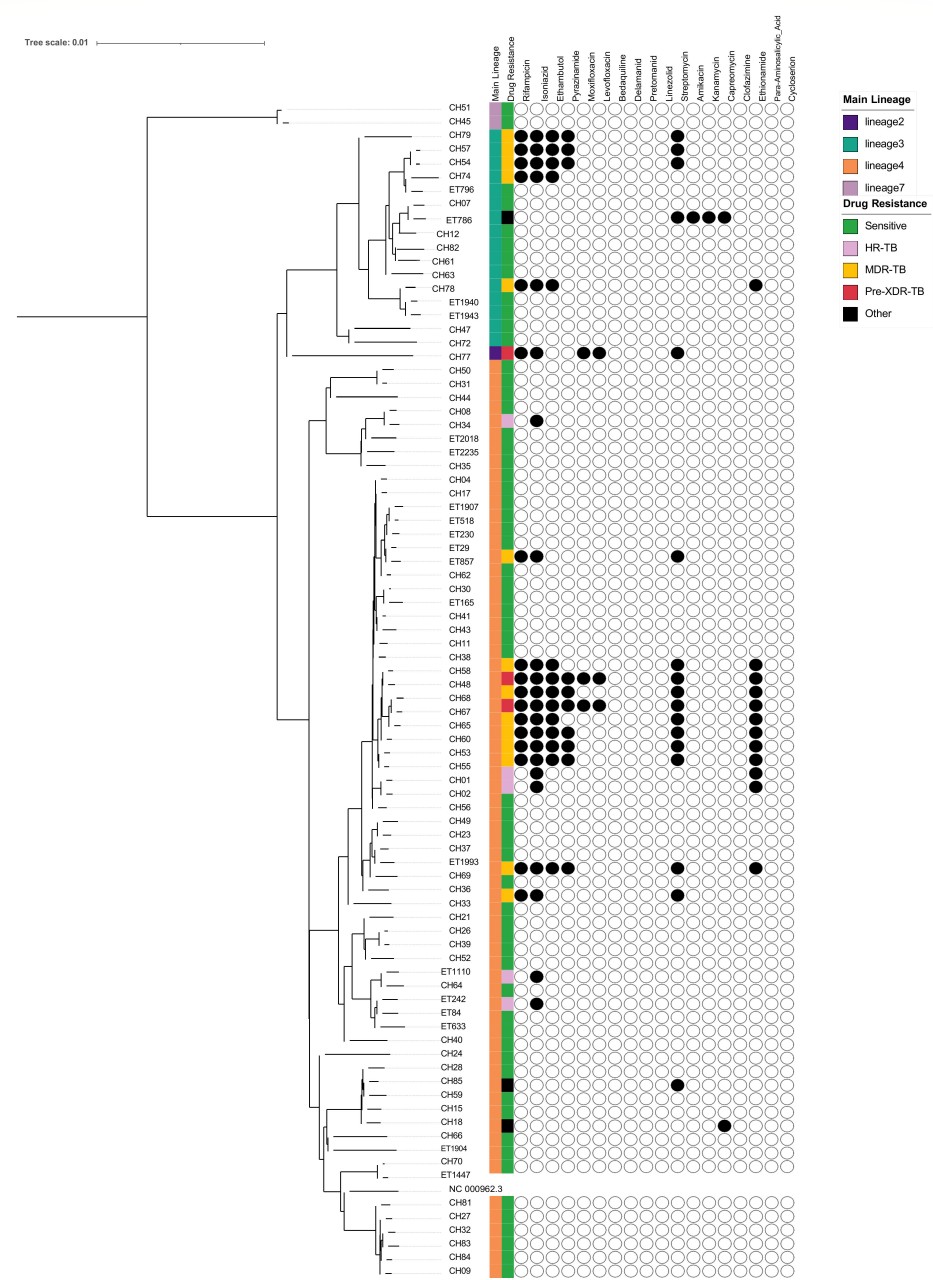

**FIG 1** Maximum likelihood phylogenetic tree generated from childhood TB isolates collected in selected health facilities in 2018–2023. The first column denotes the lineages, and the second column indicates the type of drug resistance. Genotypic resistance to first- and second-line antituberculosis drugs is represented by filled circles (presence of resistance) or empty circles (absence of resistance).

from these regions. In Oromia, sub-lineage 4.2.2.2 was predominant, representing 22/49 (44.9%) of the cases, whereas sub-lineage 4.2.1 was isolated specifically from SNNP, only from rural areas, at a rate of 10/26 (38%). Interestingly, sub-lineages 4.1.1.3, 4.2.1, 4.2.2.1, and 4.1.4 were not identified in any other regions of Ethiopia, except for these two regions (Table 2).

The geographic distribution of the identified MTB lineages and sub-lineages is shown in Fig. 3A. These lineages and sub-lineages were further analyzed by their drug susceptibility and resistance distribution (Fig. 3B). The Global Moran's I test revealed

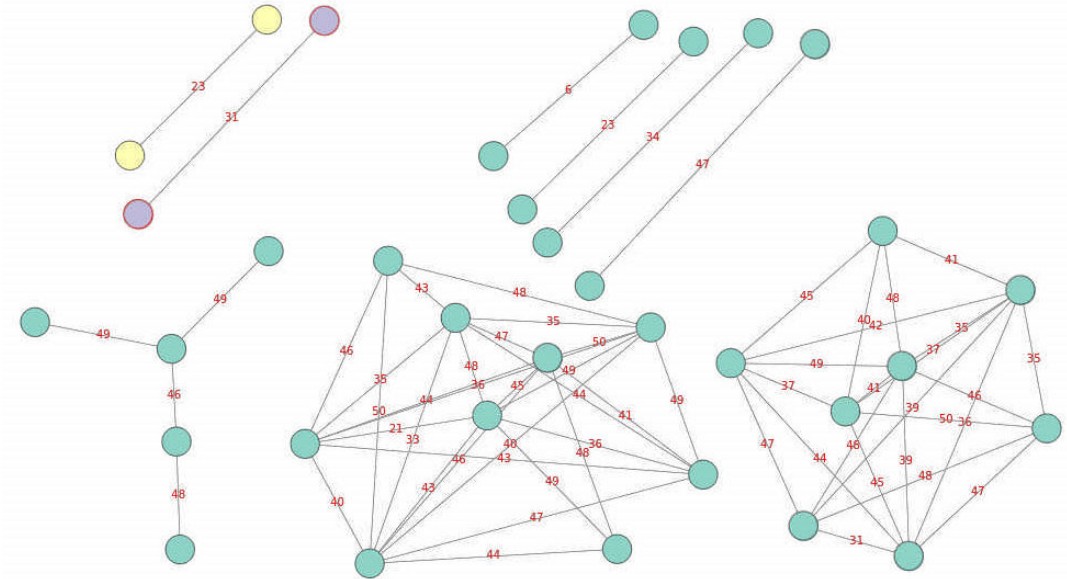

**FIG 2** The minimum spinning tree with pairwise genetic distance less than 50. Purple indicates lineage 7; yellow indicates lineage 3; and green indicates lineage 4.

that L-4 and L-4.2.2 distribution variability was statistically significant (Table 3). The hotspot analysis for L-4 is presented in Fig. 3C, highlighting significant hotspots within the pipeline network, marked by colored lines. These hotspots were primarily concentrated in Oromia and SNNP. Notably, hotspots with a 90% CI were identified in Silti and Hosaina. In contrast, the hotspots at 99% CI were found in Shashemene, Chuko, Wendo Genet, Kofele, Busa, Yirga Alem, Kebalenka, Bule, Aroresa, and Wenago. Cold spots were identified in Jimma, Becho, Guliso, and Dale at 90% CI; Bandiro, Gattira, and Limmu Kosa at 99% CI; and Ambo at 95% CI. Figure 3D shows the output specifically for sub-lineage 4.2.2.2 with significant hotspots at 99% CI in Robi, Iteya, Dera, and Adama within Oromia. Cold spots were specifically identified in Agere Mariyam, Gattira, and Limmu Kosa at 90% CI for this sub-lineage.

Among the 85 study participants, 25 were from urban areas and 60 were from rural areas. In rural areas, L-4 took the highest proportion (47/60, 80%) while L-3 was the next most abundant lineage (10/60, 16.7%). Likewise, lineages/sub-lineages L-4.2.1, L-4.8, and L-7 were only identified in rural areas, with proportions of 10/60 (16.6%), 6/60 (10%), and 2/60 (3.3%), respectively (Table 2).

**TABLE 2** National and regional geographic distribution of TB strains among children in Ethiopia[a]

| Region (n) | L-2 | L-3 | L-4.1.1 | L-4.1.2.1 | L-4.1.4 | L-4.2.1 | L-4.2.2.2 | L-4.4.1 | L-4.6 | L-4.7 | L-4.8 | L-7 |
|---|---|---|---|---|---|---|---|---|---|---|---|---|
| Tigray (1) | 0 | 0 | 0 | 1 | 0 | 0 | 0 | 0 | 0 | 0 | 0 | 0 |
| Amhara (1) | 0 | 0 | 0 | 0 | 0 | 0 | 1 | 0 | 0 | 0 | 0 | 0 |
| Oromia (49) | 1 | 10 | 0 | 3 | 1 | 0 | 22 | 0 | 5 | 1 | 4 | 2 |
| SNNP (26) | 0 | 1 | 2 | 1 | 0 | 10 | 6 | 1 | 2 | 0 | 2 | 0 |
| Gambella (2) | 0 | 2 | 0 | 0 | 0 | 0 | 0 | 0 | 0 | 0 | 0 | 0 |
| Dire Dawa (1) | 0 | 1 | 0 | 0 | 0 | 0 | 0 | 0 | 0 | 0 | 0 | 0 |
| Addis Ababa (6) | 0 | 2 | 0 | 0 | 0 | 0 | 3 | 0 | 0 | 1 | 0 | 0 |
| Rural (60) | 0 | 10 | 1 | 4 | 1 | 10 | 19 | 1 | 5 | 1 | 6 | 2 |
| Urban (25) | 1 | 6 | 1 | 1 | 0 | 0 | 13 | 0 | 2 | 1 | 0 | 0 |
| Total national (85) | 1 | 16 | 2 | 5 | 1 | 10 | 32 | 1 | 7 | 2 | 6 | 2 |

[a]L, lineage; n, number of TB strain; SNNP, Southern Nations, Nationalities, and Peoples' Region.

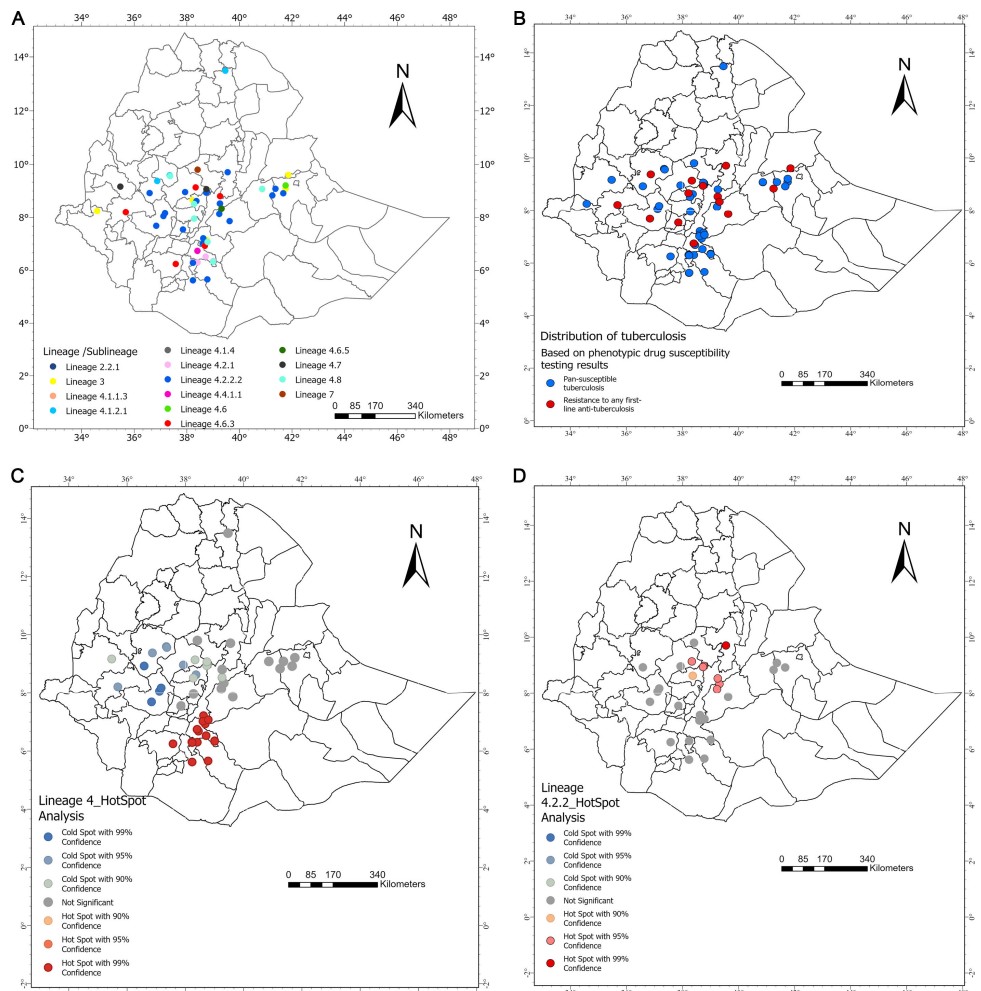

**FIG 3** (A) The distribution of tuberculosis lineages/sub-linages across Ethiopia. (B) The distribution of tuberculosis drug susceptibility and resistance. (C and D) The hotspot areas for lineages 4 and 4.2.2, respectively. The maps were created using ArcGIS.

## Drug resistance and associated mutations among the MTB lineages

The drug resistance profiles of the lineages were assessed using the TB-Profiler prediction platform (Fig. 2). Out of the 85 isolates, 25/85 (29.4%) were DR-TB, where 17/25 (68%) of the DR-TB isolates or 17/85 (20%) of the total isolates were MDR-TB/ pre-XDR-TB (Fig. 1). MDR-TB/pre-XDR-TB was mainly observed within L-3 and L-4.2.2.2, with proportions of 5/17 (29%) and 11/17 (64.7 %), respectively.

The study also identified resistance to second-line drugs. Specifically, resistance to injectable drugs was found in 2/17 (11.7%), and pre-extensively drug-resistant TB (Pre-XDR-TB) was detected in 3/17 (17.6%). These pre-XDR-TB (fluoroquinolone-resistant) strains were identified in one case from L-2 and two cases from L-4.2.2.2 (Fig. 1).

Taking phenotypic DST as the gold standard, we compared first- and second-line antituberculosis drug resistance with the corresponding genetic markers for antituberculosis drug resistance using data from WGS. Out of the total 85 TB isolates subjected to genomic and phenotypic drug resistance analysis, 26 cases (30.5%) were identified as DR-TB to at least one of the five first-line anti-TB drugs tested by phenotypic drug susceptibility testing (pDST). Among these 26 pDR-TB cases, 23 (88.5%) were concordant with WGS's anti-TB drug resistance prediction. Moreover, pDR-TB test and WGS on MDR-TB cases showed a concordance rate of 17/18 (94.4%). All WGS-identified pre-XDR-TB cases were 100% concordant with the phenotypic DST profile for fluoroquinolones.

**TABLE 3** Global spatial autocorrelation results of lineages 4 and 4.2.2

| Lineages/sub-lineages | Moran's I | z-score | P value[a] |
|---|---|---|---|
| Lineage 4 | 0.649182 | 4.561170 | **0.00005** |
| Lineage 4.2.2 | 0.649182 | 4.561170 | **0.00005** |

[a]The boldfaced P values indicate statistical significance.

Mutations detected by WGS-based predictions in genes associated with phenotypic drug resistance are listed in Table 4. The most frequent mutation associated with rifampicin resistance was p.Ser450Leu (15/17, 88.2%) in the *rpoB* gene. In two MDR samples, resistance to rifampicin was driven by distinct double mutations: one sample with two primary heteroresistance mutations, p.His445Arg and p.His445Tyr at *rpoB*, and another sample with a primary resistance mutation, p.Ser450Leu at *rpoB*, and a compensatory mutation, p.Asp747Ala at *rpoC*. Isoniazid resistance was also associated with the *ahpC*, *inhA*, and *katG* genes, the most frequent resistance mutation being p.Ser315Thr (18/21, 86%). In addition, primary resistance mutations p.Ser315Asn and p.Ser140Asn in *the katG* gene and compensatory mutation p.Ser94Ala in the *inhA* gene were identified. Variable mutational frequencies were associated with isolates that were phenotypically resistant to ethambutol and pyrazinamide. Moreover, mutations associated with resistance to second-line antituberculosis drugs included those in the *gyrA* gene, which are associated with fluoroquinolone resistance: p.Ala90Val (1/3, 33.3%), p.Asp94Asn (1/3, 33.3%), and p.Asp94Gly (1/3, 33.3%). No mutation was detected in the genes encoding resistance to delamanid, bedaquiline, and clofazimine (Table 4). Finally, chi-square/Fisher's exact test identified a history of TB contact, residence, and lineages as being statistically significantly associated with MDR-TB occurrence ($P < 0.05$) (Table 5).

## DISCUSSION

One of the study objectives was to investigate the genetic diversity of MTB collected from children with pulmonary TB by using WGS. Our findings showed the presence

**TABLE 4** Mutations detected in genes associated with phenotypic drug resistance across the 25 MTB isolates included in the study

| Phenotypic resistance (n) | Resistance gene detected with WGS (n) | Mutation (nucleotide or amino acid) change (n) |
|---|---|---|
| Rifampicin (17) | *rpoB* | p.His445Arg+p.His445Tyr (1), p.His445Cys (1), p.Ser450Leu (15) |
| | *rpoC* | p.Asp747Ala (1) |
| Isoniazid (21) | *Ahpc* | c.-81C>T (1)[a] |
| | *inhA* | p.Ser94Ala (1) |
| | *katG* | c.45_46insA (1)[a], p.Ser315Thr (18), p.Ser140Asn (1)[a], p.Ser315Asn (1) |
| Ethambutol (9) | *embA* | c.-16C>T (1) |
| | *embB* | p.Met306Ile (5), p.Asp1024Asn (1)[a], p.Gly406Ala (1), p.Asp328Gly (2)[a], p.Asp354Ala (1), p.Gly406Asp (1), p.Met306Val (1) |
| Pyrazinamide (7) | *pncA* | c.-11A>G (2)[a], c.192_193insA (1), p.Ala134Val (1), p.Asp12Ala (1), p.Trp68Gly (1) |
| Streptomycin (11) | *rpsL* | Arg86Pro (1), p.Lys88Thr (1), p.Lys43Arg (2), p.Lys88Arg (2) |
| | *gid* | c.102delG (2), p.Gly69Asp (6) |
| | *rrs* | n.906A>G (1), n.514A>C (1) |
| Fluoroquinolones (3) | *gyrA* | p.Ala90Val (1), p.Asp94Asn (1), p.Asp94Gly (1) |
| Second-line injectable (amikacin) (1) | *rrs* | n.1484G>T (1) |

[a]Indicates absence from the WHO drug resistance catalog.

**TABLE 5** MDR-TB-associated risk factors among children with TB[c]

| Variable | | Resistance profile | | P value[d] |
|---|---|---|---|---|
| | | MDR, n (%) | Susceptible, n (%) | |
| Age | 0–4 | 2 (11.1) | 6 (9) | 0.839[a] |
| | 5–9 | 2 (11.1) | 11 (16.4) | |
| | 10–14 | 14 (77.8) | 50 (74.6) | |
| Sex | Male | 8 (44.4) | 33 (49.3) | 0.717[a] |
| | Female | 10 (55.6) | 34 (50.7) | |
| Previous history | New | 14 (77.8) | 62 (92.5) | 0.071[b] |
| | Previously treated | 4 (22.2) | 5 (7.5) | |
| HIV Status | Yes | 0 | 5 (7.5) | 0.232[b] |
| | No | 18 | 62 (92.5) | |
| Had TB contact | Yes | 15 (83.3) | 35 (52.2) | **0.001**[b] |
| | No | 0 | 30 (44.8) | |
| | Unknown | 3 (16.7) | 2 (3.0) | |
| Residence | Rural | 8 (44.4) | 52 (77.6) | **0.006**[a] |
| | Urban | 10 (55.6) | 15 (22.4) | |
| Lineage | Lineage 3 | 5 (27.8) | 11 (16.4) | **0.007**[b] |
| | Lineage 4.2.2.2 | 11 (61.1) | 21 (31.3) | |
| | Other lineages | 2 (11.1) | 35 (52.2) | |

[a]P value based on chi-square test.
[b]P value based on Fisher's exact test.
[c]MDR-TB, multidrug-resistant tuberculosis; other lineages included lineages 2, 7, and 4, other than lineage 4.2.2.2, which was identified in this study.
[d]The boldfaced P values indicate statistical significance.

of genetic diversity among the TB isolates in the country, with the most prevalent tuberculosis lineage detected being L-4 (77.7%) and the lowest being L-2 (1.2%). This finding was similar to reports from previous studies on children (18, 19) and several others on adults in the country, implying that the TB lineage distribution among children in Ethiopia is comparable to that of the general population (20–25). This study is also supported by studies conducted in Russia and China, which have shown that children reflect the general situation regarding TB lineages (26). However, a household contact study in Peru showed that there was lineage-specific childhood TB where Beijing strains were more transmissible in children than were non-Beijing strains (27). This difference may be due to variations in the burden of TB strains in different countries. In Ethiopia, the prevalent linkages/sub-linkages may not have a specific target age group, unlike the observed case of Beijing strains from the previous Peru study (27).

Among the lineage/sub-lineages identified in our study, L-4.2.2.2 was the most dominant one, followed by L-3. This result aligns with the findings of several previous studies in Ethiopia (20–25), indicating that these lineages/sub-lineages have been in ongoing transmission, as children are often infected by strains prevalent within their household or local community (8, 10, 11). More importantly, the emergence of these dominant lineages/sub-lineages documented in this and previous studies calls for careful investigation into their evolution, virulence properties, and drug resistance mechanisms, as the design for appropriate intervention strategies requires inputs from such studies.

In this study, high genetic diversity was detected in the Oromia and SNNP, mainly from the L-4 lineage, as was reported from a previous study (25), showing that L-4 was more dominant than other lineages, such as L-3. Whereas L-4.2.1 was identified explicitly within the SNNP, the other Euro-American strains (L-4.4.1, L-4.6, L-4.7, and L-4.8) were found in different parts of the country, albeit rarely. However, most previous studies reported Euro-American strains as T family rather than lineages/sub-lineages, as the studies were conducted using MIRU-VNTR/spoligotyping, which is difficult to classify by lineages/sub-lineages using these latter techniques (28).

In our study, lineage 7 was identified from two patients in Wuchale Wereda, northern Shewa of Ormia. In a recent report, lineage 7, whose resulting disease is known to

progress at a slower rate than other lineages, was reported from different parts of Ethiopia. However, it was initially restricted to the Amhara region (25). Therefore, this lineage deserves close attention for enhanced surveillance and further investigation. Moreover, given the epidemiological link observed between these two cases in this lineage, it may be surprising to find as big as 31 SNP difference, which is far away from the criterion of SNP distance of 12 for a common transmission source (29), which may challenge the latter criterion and call for more research to understand if local situation may need to be considered in using the criterion of 12 SNP distance used for determining a common transmission source.

In this study, MDR-TB cases were more frequently observed among isolates belonging to lineages 3 and 4.2.2.2. Our finding is similar to the previous study (30, 31). These lineages were the predominantly prevalent lineages circulating in Ethiopia as observed from our study population and previous studies (20–25, 30, 31). Thus, MDR-TB among these lineages is more likely to reflect underlying lineage prevalence rather than intrinsic differences in drug resistance. However, further evolutionary analysis is needed to understand why these two lineages are more linked to MDR than the other lineages.

WGS provides comprehensive drug resistance profiles and associated mutations, in addition to enabling precise differentiation between strains (32). In this study, WGS in general showed a high level of concordance with pDST in providing an accurate prediction of drug susceptibility testing of anti-TB drugs. The findings agree with previous studies, which showed the utility of WGS in drug resistance surveillance and for determining therapeutic schemes (33, 34). Additionally, WGS had a short turnaround time to deliver results within 2–3 days, compared to the several weeks needed for pDST (35). Moreover, although pDST remains the reference standard for detecting drug resistance, WGS enables the simultaneous identification of resistance-associated mutations to both first- and second-line drugs across the genome (33–35). We recommend further research into the feasibility of using direct detection of drug resistance markers from sputum samples using WGS under the Ethiopian setting, as this could eliminate the need for culture, which is particularly challenging in children's cases (36).

In our study, the most frequently detected mutations associated with rifampicin and isoniazid resistance were Ser450Leu (15/17, 88.2%) in *the rpoB* gene and p.Ser315Thr (18/21, 86%) in the *kat* gene, respectively. These levels of resistance mutations were also reported from another previous study (37). Furthermore, in this study, a heteroresistant isolate was found to harbor two distinct mutations in the *rpoB* gene, namely, p.His445Arg and p.His445Tyr, both of which occur at the same codon but result in different amino acid substitutions (arginine and tyrosine, respectively). These mutations are located within the rifampicin resistance-determining region of the *rpoB* gene, which is known to be the primary target for rifampicin resistance, strongly suggesting that this isolate has developed a high level of resistance to rifampicin (38). The other mutation in the *rpoB* gene at p.Ser450Leu simultaneously carries a mutation in the *rpoC* gene at p.Asp747Ala. The *rpoC* gene encodes another sub-unit of RNA polymerase, and mutations in *rpoC* are often seen alongside *rpoB* mutations in rifampicin-resistant strains such as the one mentioned above (39). In this simultaneous occurrence of mutations in the two genes, the *rpoC* mutation may serve as a compensatory mechanism, helping to mitigate any fitness costs associated with the *rpoB* mutation, allowing the bacterium to survive and replicate more effectively despite the resistance mutation (39).

Finally, we acknowledge that our study has some limitations. Firstly, several TB cultures could not be reactivated in Mycobacteria Growth Indicator Tube (MGIT) medium, despite multiple attempts. Secondly, due to budget limitations, we were unable to repeat sequencing for culture-positive isolates that failed during the initial sequencing run. Moreover, not enough cases were identified in some study sites, such as the Amhara and Tigray regions, to fully understand if there is a difference in lineage distribution between regions. We also have not observed the cost-effectiveness, turnaround time under routine programmatic conditions, and implementation feasibility of WGS.

In conclusion, the findings underscore the high genetic diversity of MTB, with a predominance of L-4, specifically L-4.2.2.2, where DR-TB was most frequently observed with this lineage. We also observed a region-specific sub-lineage (L-4.2.1) in the SNNP, most notably in rural children. The study also highlighted the importance of WGS for a comprehensive analysis of drug resistance mutations (first- and second-line drugs) and associated mutations. Therefore, the study lays the groundwork for integrating genomic surveillance into the control of childhood TB and drug resistance. However, further feasibility, implementation, and cost-effectiveness studies are needed to evaluate whole-genome sequencing and other alternative next-generation sequencing methods (targeted sequencing approaches, including Nanopore-based methods) that enable resistance detection directly from sputum and may help address challenges related to culturing childhood TB samples. Moreover, we also recommend contact tracing for a better understanding of transmission dynamics between children and adults.

## MATERIALS AND METHODS

### Study area

This study was conducted in 62 selected TB diagnostic health facilities across Ethiopia between August 2017 and January 2018. Of these 62 health facilities, data collection continued for 36 health facilities until 2023. Ethiopia, located in the northeastern part of Africa, comprises 2 city administrations and 12 administrative regions. Our study was conducted in the two administrative cities and five regions of Ethiopia.

### Study design

This study was conducted using a retrospective cross-sectional study design.

### Study population

The study included children under 15 years of age who were diagnosed with bacteriologically confirmed pulmonary TB or DR-TB between January 2017 and June 2023.

### Sample size and sampling technique

We calculated the sample size using a single population proportion formula as follows:

$$n = \frac{Z_{\alpha-1/2}p(1-p)}{d^2},$$

where $n$ is the required sample size; in $Z_{\alpha-1/2}$, $Z$ is the critical point for 95% confidence level under a normal distribution ($Z = 1.96$); $p$ is the true population proportion of lineages; and $d$ is the level of precision = 0.05. The true population proportion of lineages is taken from the previous study, where the proportion of lineage 1 (Indo-Oceanic) was 6.5% (40). After substituting the appropriate values into the above formula, the sample size became 94. Assuming a 10% loss of viability or contamination rate and registration-related errors, we planned to include 104 individuals with TB isolates. However, we did not restrain ourselves to this sample size; we included all identified isolates regardless of the sample size estimation.

### Sampling technique

All children with bacteriologically confirmed pulmonary TB at the selected health facilities were included in the study until the desired sample size was achieved, where only a total of 102 children with TB were identified through the third round of drug resistance survey from 2017 to 2018 data collection time (41), of whom 88 were culture positive, whose isolates were stored at −80°C. However, only 75 of these 88 archived isolates were recovered from the sub-culturing. Additionally, 35 culture-confirmed TB

cases were included from MDR-TB treatment initiative centers that continued data collection until 2023; however, recovery efforts failed for five of these isolates. Therefore, a total of 105 culture-confirmed tuberculosis cases were recovered, of which good DNA quality and quantity were achieved for 96 isolates, and these were subsequently subjected to genomic analysis (Fig. 4).

## Sample collection and TB culture processing

Sputum or gastric aspirate samples for TB culture processing were collected from bacteriologically confirmed pulmonary TB patients and tested by smear or GeneXpert MTB/RIF, as previously reported in a published paper (4). The samples were processed for TB culture by decontaminating the sputum with an equal volume of NALC-sodium hydroxide solution for 15 minutes, followed by centrifugation at $3,000 \times g$. The sediment was then re-suspended with 2 mL of phosphate buffer and inoculated into an MGIT (Becton Dickinson, Sparks, MD, USA). The MGIT was incubated in the MGIT machine, which flagged the results as negative or positive. Positive MGIT cultures were tested on blood agar and subjected to Ziehl-Neelsen staining to check for contaminants. Blood agar-negative, smear-positive samples were further confirmed by rapid immunochromatographic assay (SD Bioline) for the presence of MTB.

## pDST

Phenotypic first- and second-line DSTs were performed using the Mycobacterium Growth Indicator Tube 960 System (BACTEC, Inc), based on WHO-recommended critical concentrations for first- and second-line anti-TB drugs (42). Drug susceptibility testing was performed using the following critical concentrations: isoniazid at 0.1 µg/mL, rifampicin at 0.5 µg/mL, streptomycin at 1.0 µg/mL, pyrazinamide at 100 µg/mL, ethambutol at 5.0 µg/mL, levofloxacin at 1.0 µg/mL, moxifloxacin at 0.25 µg/mL, amikacin at 1.0 µg/mL, delamanid at 0.06 µg/mL, bedaquiline at 1.0 µg/mL, clofazimine at 1.0 µg/mL, and linezolid at 1.0 µg/mL and interpreted by comparing the growth with a drug-free control of the same specimen.

## DNA extraction and whole-genome sequencing

MTB isolates were sub-cultured again in MGIT, and genomic DNA extraction was performed using the cetyltrimethylammonium bromide-lysozyme method as described in a previously published protocol (43). Quality control for the extracted DNA was conducted using agarose gel electrophoresis, Qubit, and NanoDrop to determine the integrity, size, concentration, and quality of the DNA. Libraries for whole-genome sequencing were prepared using the Nextera XT Library Prep Kit according to the manufacturer's instructions (Illumina, Little Chesterford, UK) and sequenced on an Illumina NextSeq 500/550 platform. Additionally, because the sequences produced from these isolates in our lab were of poor quality, we utilized 20 data sets available in the form of FastQ produced from childhood TB isolates obtained from the same targeted study sites, which were already sequenced as part of the third round of drug resistance surveys using the Illumina NextSeq 500 platform in Milan (Fig. 4) (44).

## Bioinformatics analysis

Initially, the quality of paired-end raw reads was assessed using standard FAST-QC tools. Trimming and filtering steps were performed by Trimmomatic version 0.33 to eliminate low-quality sequences, adapter contamination, and sequences at the start and end of reads (45). Duplicate reads were marked to control for misalignment. High-quality paired-end reads with a Phred quality score above 20 and a minimum length of 50 base pairs were retained for downstream processing. Species confirmation and contamination checking were carried out using Kmer-based taxonomy classification with Kraken versions 1 and 2 (46). Alignment was conducted by mapping to the *M. tuberculosis* H37Rv

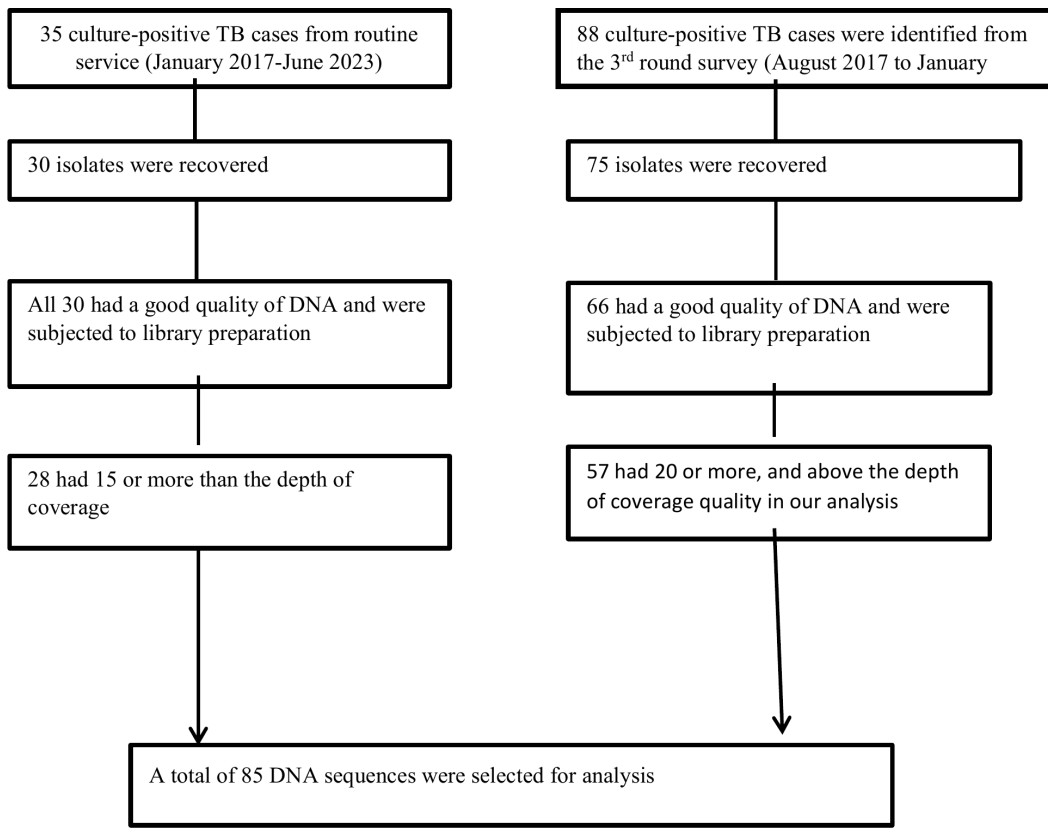

NB: Twenty isolates were obtained from a sequence done for the 3rd round drug resistance survey

**FIG 4** Workflow of data collection and processing.

reference genome (GenBank accession number NC_000962.3) using BWA-MEM version 0.7.16, and mapping quality was assessed using Samtools stats (47) to ensure that the average depth of coverage was greater than 15× and genome coverage exceeded 99%.

## Phylogenetic study and lineage assignment

The variants were called with bcftools, and the SNPs were filtered specifically for the construction of phylogenetic trees. The positions of the SNPs within PE/PPE or any other repetitive regions of low mapping ability score were removed from the final core-genome alignment. An SNP-based phylogenetic tree was constructed using IQ-TREE maximum-likelihood models with 1,000 bootstrap replications (48). Tree visualization and annotation were performed using the iTOL: Interactive Tree of Life web server (49). MTB lineage assignment was conducted using TB-Profiler version 6.2.0 (50).

## Determination of drug resistance

To determine the genotypic drug resistance profile, TB-Profiler command-line pipelines (TB-Profiler version 6.2.0) were used to identify antibiotic resistance markers (SNPs, indels, and mutations) (50). Drug-resistance profiles identified by WGS were compared with pDST, and any discordant results were re-evaluated using repeat pDST to confirm discrepancies.

## Genomic relatedness cluster analysis

Pairwise SNP distances were calculated among all sequences using the snp-dists 0.7.0 tool (51). Based on the similarity and dissimilarity matrix data set, the cluster association

was predicted among each isolate. The maximum linkage thresholds of genomic relatedness or cluster in this study were set as 50 SNP distances according to the previous study used by Coll et al. (52). Visualization of cluster networks and threshold-based minimum spanning tree was generated using the GraphSNP interactive source code (53).

## Spatial analysis

GIS tools were employed to analyze pipeline incident hotspots. Global spatial patterns were evaluated using spatial autocorrelation techniques (i.e., Global Moran's I) to assess the clustering, dispersion, or random distribution of incidents (54). A buffer zone of 5 m was established for hotspot analysis around the pipe network. Geographic data were plotted using GIS to show the locations of residential clusters at the *woreda/kebele* level.

## Statistical analysis

Statistical analysis was performed using SPSS 22.0 (SPSS Inc., Chicago, IL, USA). Descriptive statistics were used to identify the sociodemographic characteristics of childhood TB. The clinical characteristics were compared using the chi-square test or Fisher's exact test and considered statistically significant at $P < 0.05$.

## ACKNOWLEDGMENTS

We acknowledge the National and Regional Reference Laboratories, Addis Ababa University, and the Armauer Hansen Research Institute in Ethiopia, Emory University, and the TB Supranational Reference Laboratory at the San Raffaele Scientific Institute in Milan for their cooperation and for facilitating the data collection and laboratory analysis. This study was partly supported by a grant from the U.S. National Institutes of Health Fogarty International Center (D43TW009127). The funder had no role in study design, data collection, interpretation, or the decision to submit the work for publication.

Conceptualization, funding acquisition, and writing (original draft): Y.A.; investigation: Y.A., G.G., A.G.W., B.Z., G.D., K.M., S.M., and D.H.A.; validation: Y.A., A.S., R.A., W.E.A., and M.A.; methodology: Y.A., R.A., W.E.A., D.H.A., A.G.W., G.G., and M.G.; performance of the bioinformatics analysis: Y.A. and A.A.; supervision: W.E.A.; writing (review and editing of the manuscript) and approval for publication: all authors.

## AUTHOR AFFILIATIONS

[1]Department of Microbiology, Immunology and Parasitology, College of Health Sciences, Addis Ababa University, Addis Ababa, Ethiopia
[2]Ethiopian Public Health Institute, Addis Ababa, Ethiopia
[3]Armauer Hansen Research Institute, Addis Ababa, Ethiopia
[4]Department of Pediatrics and Child Health, College of Health Sciences, Addis Ababa University, Addis Ababa, Ethiopia
[5]Department of Medicine, School of Medicine, Emory University, Atlanta, Georgia, USA

## AUTHOR ORCIDs

Yeshiwork Abebaw ⓘ http://orcid.org/0000-0001-6039-3430
Gebremedhin Gebremicael ⓘ http://orcid.org/0000-0001-6726-9600
Getu Diriba ⓘ http://orcid.org/0000-0002-0166-682X
Muluwork Getahun ⓘ http://orcid.org/0000-0003-3407-983X

## FUNDING

| Funder | Grant(s) | Author(s) |
| --- | --- | --- |
| Fogarty International Center | D43TW009127 | Yeshiwork Abebaw |

## AUTHOR CONTRIBUTIONS

Yeshiwork Abebaw, Conceptualization, Data curation, Formal analysis, Funding acquisition, Investigation, Methodology, Resources, Supervision, Validation, Visualization, Writing – original draft, Writing – review and editing | Abaysew Ayele, Data curation, Formal analysis, Methodology, Writing – review and editing | Dawit Hailu Alemayehu, Investigation, Methodology, Writing – review and editing | Gebremedhin Gebremicael, Investigation, Methodology, Writing – review and editing | Getu Diriba, Investigation, Writing – review and editing | Atsbeha Gebreegziabxier Weldemariam, Investigation, Methodology, Writing – review and editing | Betselot Zerihun, Investigation, Writing – review and editing | Shewki Moga, Investigation, Writing – review and editing | Bethlehem Adnew, Formal analysis, Writing – review and editing | Kalkidan Melaku, Investigation, Writing – review and editing | Muluwork Getahun, Methodology, Writing – review and editing | Rahel Argaw, Methodology, Validation, Writing – review and editing | Markos Abebe, Methodology, Validation, Writing – review and editing | Woldaregay Erku Abegaz, Data curation, Methodology, Validation, Visualization, Writing – review and editing.

## DATA AVAILABILITY

All the information related to the manuscript is included in this paper, and the additional information is provided in the Miscellaneous File. WGS raw reads were submitted to NCBI as FASTQ files under study accession numbers PRJNA1104194 and PRJNA1204469.

## ETHICS APPROVAL

Ethical approval was obtained from the Departmental Research and Ethics Review Committee of the Department of Microbiology, Immunology & Parasitology (DMIP) and IRB of the College of Health Sciences, Addis Ababa University (reference number 103/20/DMIP), and the Ethiopian Public Health Institute (EPHI) scientific and ethical review committee (reference number EPHI-IRB-239-2020). This study included both stored childhood TB isolates and TB isolates collected directly from study participants. Therefore, informed consent was waived for stored childhood TB isolates by both Addis Ababa University IRB and the EPHI ethical review committee. However, for sample collection from direct study participants, the parents or guardians of all eligible individuals were informed about the study, and their consent was obtained. Additionally, assent was sought from children aged over 12 years. All methods were performed in accordance with relevant guidelines and regulations.

## ADDITIONAL FILES

The following material is available online.

### Open Peer Review

**PEER REVIEW HISTORY (review-history.pdf).** An accounting of the reviewer comments and feedback.

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
