## [Reviewer comments · Microbiology Spectrum]

Microbiology Spectrum

Genetic Diversity and Drug Resistance Profiles of *Mycobacterium tuberculosis* among Ethiopian Children as Determined by Whole-Genome Sequencing

Yeshiwork Abebaw, Abaysew Ayele, Dawit Alemayehu, Gebremedhin Gebremicael, Getu Diriba, Atsbeha Weldemariam, Betselot Zerihun, Shewki Moga, Bethlehem Adnew, Kalkidan Melaku, Muluwork Getahun, Rahel Argaw, Anandi Sheth, Markos Abebe, and Woldaregay Abegaz

Corresponding Author(s): Yeshiwork Abebaw, Addis Ababa University School of Medicine

Review Timeline:

Submission Date:	September 4, 2025
Editorial Decision:	October 26, 2025
Revision Received:	November 26, 2025
Editorial Decision:	December 19, 2025
Revision Received:	January 1, 2026
Accepted:	January 6, 2026

Editor: Florence Doucet-Populaire

Reviewer(s): Disclosure of reviewer identity is with reference to reviewer comments included in decision letter(s). The following individuals involved in review of your submission have agreed to reveal their identity: Hamza Babiker (Reviewer #2); Hao Li (Reviewer #3)

Transaction Report:

DOI: <https://doi.org/10.1128/spectrum.02736-25>

Re: Spectrum02736-25 (**Population structure of childhood TB: Phylogenomic, Phylogeography, and Drug-Resistance pattern of *Mycobacterium Tuberculosis* isolates in Ethiopia**)

Dear Mrs. Yeshiwork Abebaw:

Thank you for the privilege of reviewing your work. Below you will find my comments, instructions from the Spectrum editorial office, and the reviewer comments.

Revision Guidelines

Sincerely,
Florence Doucet-Populaire
Editor
Microbiology Spectrum

Reviewer #1 (Comments for the Author):

The current manuscript describes phylogenomic, geospatial, and drug resistance characteristics of childhood tuberculosis (TB) in Ethiopia. As the authors note, previous studies of TB in Ethiopia have focused on adults and/or used molecular techniques (e.g., spoligotyping) with limited discriminatory power.

The primary limitation of this study is sample size. Considering the high burden of pediatric TB in Ethiopia, it's unclear why

sampling from >36 health facilities for >6 years (2017-2023) uncovered only 123 culture positive cases and 85 sequences. The authors do acknowledge that "not enough cases were identified in some of the study sites" [lines 294-295]. Even so, some results are described as 'statistically significant' even though the 'Sample size' calculations [lines 319-329] suggest that the study is underpowered. In particular, the sub-lineage analyses (e.g., Table 2) are based too few strains for 'significant' conclusions. Note, the 'Sample size' section states: " the true population proportion of lineages is taken from the previous study, where the proportion of L-1 was 6.5% (33)". However, ref 33 only includes spoligotyping data. How was the L-1 percent derived from SIT designations?

The authors comment that "Drug-resistant tuberculosis in children is typically transmitted through prolonged close contact with an adult who has DR-TB [lines 71-72] and "children are often infected by strains that are prevalent within the household or local community" [lines 235-236]. It appears that several of the co-authors have also published genomic epidemiology studies of adults (e.g., Moga et al. 2025. *Open Forum Infect Dis.* 12(7):ofaf367. doi: 10.1093/ofid/ofaf367; Mollalign et al. 2025. *IJTLD* 29(8):355-361. doi: 10.5588/ijtld.24.0685). Do those datasets include WGS results for any relatives or close contacts of the children in this study? A sub-analysis of (potential) TB transmission within families or households would be interesting. Consider revising the description of drug-resistance mutations to distinguish nucleotide and amino acid changes. For example, lines 280-282 state: "mutations... in the rpoB gene, namely, p.His445Arg and p.His445Tyr, both occur at the same position (His445) but result in different amino acid substitutions". The mutations aren't at the same position. His>Tyr involves the first position of the codon (i.e., CAT>TAT or CAC>TAC) whereas the His>Arg involves the second position (i.e., CAT>CGT or CAC>CGC).

Reviewer #2 (Comments for the Author):

The manuscript reports WGS-based analysis of *Mycobacterium tuberculosis* isolates, describing genetic diversity, lineage distribution, and drug-resistance profiles. The topic is relevant to TB control and genomic surveillance in Ethiopia. However, the manuscript needs major editing for clearer study aim/justification, better organization to remove redundancy.

Major concerns

1. Study Objective and rationale. The aim to assess DR-TB in children and compare with adults, is stated but not tested, as no adult was included for comparison. The rationale and research question should be explicitly defined.
2. Organization and redundancy. Results are fragmented and repetitive. Key findings are obscured by repeated presentation of similar numerical data in different sections (e.g lineages distribution and drug resistance profile)
3. Data Interpretation. Many statements are descriptive and lack statistical support. Include tests (e.g., chi-square, Fst, or AMOVA) to substantiate claims of "significant diversity" or lineage-resistance associations. For example, Line 138: The statement claiming "significant genetic diversity".

Minor points

1. Title: Should edited to clearly reflect study aim and data presented.
Suggested: "Diversity and Drug Resistance Patterns of *Mycobacterium tuberculosis* Isolates Infecting Children in Ethiopia."
2. Line 105, "Population structure of *M. tuberculosis* lineage and genotype drug resistance based on WGS prediction" replace with "Diversity of *M. tuberculosis* lineage and drug resistance prediction based on WGS"
3. Line 136. "Geographic Distribution of the detected TB strains" replace with "Geographic Distribution of MTB strains"
4. Line 136: "Geographic Distribution of MTB Strains."
5. Supplementary Tables 2-3: Merge due to overlap.
6. Figure 4A: Redraw for clearer depiction of sub-lineage prevalence.
7. Grammar: Line 150, change "this lineage" to "these lineages."
8. Edit Figure 1 to clarify the time of collection for different sets of MTB isolates and DNA sequence used in the study.

Title: Genetic Diversity and Drug Resistance Profile of Mycobacterium tuberculosis among Ethiopian Children as Determined by Whole-Genome Sequencing.

Dear Editor,

Prof. Florence Doucet-Populaire

We would like to express our sincere gratitude for the opportunity to revise and resubmit our manuscript in response to the reviewers' comments. We also thank the reviewers for their valuable and insightful feedback, which has helped us to improve the quality and clarity of our work. Below, we provide a detailed point-by-point response to each reviewer's comment. In the revised manuscript, all changes made in response to the reviewers' suggestions are highlighted in track changes for ease of reference.

Reviewer 1 comments:

Comment 1

The primary limitation of this study is sample size. Considering the high burden of pediatric TB in Ethiopia, it's unclear why sampling from >36 health facilities for >6 years (2017-2023) uncovered only 123 culture-positive cases and 85 sequences.

Response to Comment 1

Thank you very much for your concern regarding the sample size. Most of the isolates used in this study were obtained from stored samples collected during the Third National Drug Resistance Survey conducted between August 2017 and January 2019. The survey included 199 health facilities representing all regions of Ethiopia and covered all age groups. However, bacteriologically confirmed childhood pulmonary TB cases were reported from only 62 out of 199 (31%) health facilities. This finding has been reported in our previous publication (Abebaw, Y. et al., 2023. BMC Pediatrics, 23, 418. <https://doi.org/10.1186/s12887-023-04240-6>).

To increase the sample size, we further included culture-positive isolates collected through routine diagnostic services between January 2017 and June 2023. Among the 123 culture-confirmed TB cases identified, 105 isolates were recovered, and 85 yielded high-quality sequencing data. Due to budget limitations, we were unable to repeat sequencing for culture-positive isolates that failed during the initial sequencing run.

We have now clarified this in the revised manuscript (lines 257-258). Additionally, while we initially presented a sample size calculation to demonstrate our initial study design, we have now specified that all available sequences were included in the analysis, regardless of the initial sample size estimation (lines 293-294).

Despite this limitation, this study represents the first genomic study of *Mycobacterium tuberculosis* isolates from Ethiopian children based on 85 whole-genome sequences, and provides important baseline data on the genetic diversity and drug resistance patterns of childhood TB in the country.

Comment 2

The authors do acknowledge that "not enough cases were identified in some of the study sites" [lines 294-295]. Even so, some results are described as 'statistically significant' even though the 'Sample size' calculations [lines 319-329] suggest that the study is underpowered. In particular, the sub-lineage analyses (e.g., Table 2) are based on too few strains for 'significant' conclusions.

Response to Comment 2

Thank you very much for this valuable comment. We agree that the sub-lineage analyses presented in Table 2 were based on a small number of strains and therefore lacked sufficient statistical power to support meaningful conclusions. Our initial intention was to explore potential associations between lineage distribution, drug resistance, and children's residence; however, given the limited sample size, we recognize that these findings may be misleading. Accordingly, we have removed the sub-lineage analysis and Table 2 from the revised manuscript and adjusted the related text to avoid over interpretation of the results.

Comment 3

Note that the 'Sample size' section states: 'The true population proportion of lineages is taken from the previous study, where the proportion of L-1 was 6.5% (33).' However, ref 33 only includes spoligotyping data. How was the L-1 percent derived from SIT designations?

Response for comment 3

Thank you very much for your observation. The percentage for Lineage 1 (Indo-Oceanic) was directly taken from the previously published report cited as reference 33. However, we acknowledge that this reference was based on spoligotyping data and that our earlier description

contained a clerical error in referring to it as Lineage 1. We have now specified the lineage 1 from which the data was taken as Lineage 1 (Indo-Oceanic) in the revised manuscript (line 291), and apologize for the confusion caused.

Comment 4

The authors comment that "Drug-resistant tuberculosis in children is typically transmitted through prolonged close contact with an adult who has DR-TB [lines 71-72] and "children are often infected by strains that are prevalent within the household or local community" [lines 235-236]. It appears that several of the co-authors have also published genomic epidemiology studies of adults (e.g., Moga et al. 2025. *Open Forum Infect Dis.* 12(7):ofaf367. doi: 10.1093/ofid/ofaf367; Mollalign et al. 2025. *IJTL D* 29(8):355-361. doi: 10.5588/ijtld.24.0685). Do those datasets include WGS results for any relatives or close contacts of the children in this study? A sub-analysis of (potential) TB transmission within families or households would be interesting.

Response for comment 4

Thank you very much for this valuable observation. Indeed, genomic epidemiology studies on tuberculosis in Ethiopia are emerging, and a sub-analysis of potential transmission within families or households would be highly informative. However, the study by Mollalign et al. (2025, *IJTL D*, 29(8):355–361, doi:10.5588/ijtld.24.0685) focused on extrapulmonary TB, which differs from our dataset of pulmonary TB in children.

As mentioned above, our dataset includes isolates collected as part of the cross-sectional study by Moga et al. (2025, *Open Forum Infectious Diseases*, 12(7):ofaf367, doi:10.1093/ofid/ofaf367). Unfortunately, that study did not include the collection of data on relatives or close household contacts of the children. Therefore, it was not possible to perform a transmission sub-analysis within families or households in the present study.

We appreciate the reviewer's suggestion and have now added this point as a recommendation for future research in the revised manuscript as follows: "We recommend contact tracing for a better understanding of transmission dynamics between children and adults" (lines 271-272).

Comment 5

Consider revising the description of drug-resistance mutations to distinguish nucleotide and amino acid changes. For example, lines 280-282 state: "mutations... in the rpoB gene, namely, p.His445Arg and p.His445Tyr, both occur at the same position (His445) but result in different amino acid substitutions". The mutations aren't at the same position. His>Tyr involves the first position of the codon (i.e., CAT>TAT or CAC>TAC) whereas the His>Arg involves the second position (i.e., CAT>CGT or CAC>CGC).

Response for comment 5

Thank you very much for this insightful comment. We agree with the reviewer's observation that the His>Tyr and His>Arg substitutions in the rpoB gene result from distinct nucleotide changes within the same codon, rather than mutations occurring at the same nucleotide position. We have revised the text accordingly in the revised manuscript (line 246-248) to distinguish between nucleotide- and amino acid-level changes clearly.

Furthermore, both mutations were identified in a Lineage 3 isolate at 67% allele frequency, supporting the presence of distinct rifampicin-resistant subpopulations within the same bacterial population (heteroresistant). Accordingly, we have updated the Results section (lines 174–176) and the Discussion (lines line 246-248) to reflect this interpretation.

Reviewer 2 comment:

Comment:

The manuscript reports WGS-based analysis of Mycobacterium tuberculosis isolates, describing genetic diversity, lineage distribution, and drug-resistance profiles. The topic is relevant to TB control and genomic surveillance in Ethiopia. However, the manuscript needs major editing for clearer study aim/justification, better organization to remove redundancy.

Response to Comment: Thank you very much for your positive feedback and constructive suggestions. We have carefully revised the manuscript to improve the clarity of the study aim, justification, and to enhance the logical flow of the text and eliminate redundancy. All relevant sections have been edited accordingly.

Major comment 1: Study Objective and rationale. The aim to assess DR-TB in children and compare with adults, is stated but not tested, as no adult was included for comparison. The rationale and research question should be explicitly defined.

Response to Major Comment 1:

Thank you very much for your comment, and we apologize for the confusion caused. Our study did not aim to compare childhood cases with adults, but rather to examine the genetic diversity and drug resistance patterns of *Mycobacterium tuberculosis* in children. We have revised the study objective and rationale in the manuscript (lines 92-99) to reflect this scope clearly and to remove any statements implying comparison with adult data.

Major comment 2. Organization and redundancy. Results are fragmented and repetitive. Key findings are obscured by repeated presentation of similar numerical data in different sections (e.g, lineage distribution and drug resistance profile).

Response to Major Comment 2

Thank you very much for raising this important comment. We agree that the Results section was previously fragmented and repetitive. In the revised version, we have restructured the Results into clear subsections: Lineage Diversity and Clustering, Geographic Distribution, and Drug Resistance and Associated Mutations among the Lineages. We have also removed redundant numerical data to improve clarity and focus on key findings.

Major Comment 3. Data Interpretation. Many statements are descriptive and lack statistical support. Include tests (e.g., chi-square, Fst, or AMOVA) to substantiate claims of "significant diversity" or lineage-resistance associations. For example, Line 138: The statement claiming "significant genetic diversity".

Response to Major Comment 3.

Thank you very much for your concern regarding data interpretation. We agree with the reviewer's observation and have revised the manuscript accordingly. Statements that lacked statistical support, such as those previously based only on descriptive data or small subgroup analyses (e.g., the former sub-lineage results in Table 2), have been removed. These sub-lineage analyses were based on a limited number of strains and therefore lacked sufficient statistical power for

meaningful conclusions. Regarding line 138, we have revised the phrase “significant genetic diversity” to “high genetic diversity,” since our findings indicate the presence of four major lineages (L2, L3, L4, and L7) and about 15 sub-lineages within Lineage 4, but without statistical testing to support significance (line 128). Additionally, we have now clarified that Chi-square or Fisher’s exact tests were applied to assess associations between drug resistance and potential risk factors, and the corresponding results are reported in the revised manuscript (lines 186-188 and Table 5).

Minor points Comment 1. Title: Should be edited to clearly reflect the study aim and data presented.

Suggested: "Diversity and Drug Resistance Patterns of Mycobacterium tuberculosis Isolates Infecting Children in Ethiopia."

Response to minor Comment 1.

Thank you very much for this helpful suggestion. We have revised the title to better reflect the study aim and the data presented. The revised title now reads: “Genetic Diversity and Drug Resistance Profile of Mycobacterium tuberculosis among Ethiopian Children as Determined by Whole-Genome Sequencing.”

Minor points Comment 2. Line 105, "Population structure of M. tuberculosis lineage and genotype drug resistance based on WGS prediction" replace with "Diversity of M. tuberculosis lineage and drug resistance prediction based on WGS"

Response to minor Comment 2: Thank you very much for your suggestion. We have accepted it and replaced the phrase with “Lineage Diversity and Clustering in line 106 and Drug Resistance and Associated Mutations among the Lineages in line 154”.

Minor points Comment 3. Line 136. "Geographic Distribution of the detected TB strains" replace with "Geographic Distribution of MTB strains"

Response to minor Comment 3: Thank you very much for your suggestion. We have accepted it and replaced the phrase with “Geographic Distribution of MTB strains” in the revised manuscript (line 126).

Minor points Comment 5. Supplementary Tables 2-3: Merge due to overlap.

Response to minor Comment 5: Thank you very much for your suggestion. We have accepted it and merged Supplementary Tables 2 and 3 into a single table, which is now presented as Table 2 in the revised manuscript. We have also removed the previous supplementary materials (Supplementary Tables 1, 2, and 3) and updated the supplementary file type to “Miscellaneous File” for the revised Table 1 and the combined table (now Table 2).

Minor points Comment 6. Figure 4A: Redraw for clearer depiction of sub-lineage prevalence

Response to minor Comment 6:

Thank you very much for your valuable observation. We have revised and redrawn Figure 4A to provide a clearer depiction of sub-lineage prevalence in the revised manuscript.

Minor points Comment 7. Grammar: Line 150, change "this lineage" to "these lineages."

Response to minor Comment 7: We have accepted the suggestion and corrected the grammatical error, changing “this lineage” to “these lineages” in the revised manuscript (line 136).

Minor points Comment 8. Edit Figure 1 to clarify the time of collection for different sets of MTB isolates and DNA sequence used in the study.

Response to minor Comment 8: We have accepted the suggestion and edited Figure 1 to clearly show the time of collection for the different sets of *M. tuberculosis* isolates and DNA sequences used in the study.

Re: Spectrum02736-25R1 (**Genetic Diversity and Drug Resistance Profiles of Mycobacterium tuberculosis among Ethiopian Children as Determined by Whole-Genome Sequencing**)

Dear Mrs. Yeshiwork Abebaw:

Thank you for the privilege of reviewing your work. Below you will find my comments, instructions from the Spectrum editorial office, and the reviewer comments.

Revision Guidelines

Sincerely,
Florence Doucet-Populaire
Editor
Microbiology Spectrum

Reviewer #1 (Comments for the Author):

In the current study, Abebaw and colleagues describe the genetic diversity and drug resistance patterns of Mycobacterium tuberculosis among Ethiopian children. This revised manuscript has an updated title and addresses several issues raised by previous reviewers. In their response to the reviewers, the authors also acknowledge that various statistical analyses were compromised by the limited number of samples available for study. Even so, the Discussion still mentions "statistically

significant associations" between MDR-TB status and specific Mtb lineages. Such trends are more easily explained by sampling biases. There are 12 lineages/sublineages described in the study, but the 2 most common (L-4.2.2.2 and L-3) represent 56% (48/85) of all isolates whereas the 6 least common represent only 10.5% (9/85). Those numbers are too small for a meaningful comparison. For lineages L-2, L-4.1.4 or L-4.4.1, a single resistant isolate would have given scores of 100% MDR. Sampling also explains the "high genetic diversity" detected in Oromia and SNNP. Those two regions accounted for >85% of all samples. There is no genetic diversity in Amhara, Dire Dawa or Tigray because each of those regions were only represented by single cases.

The authors could consider re-framing their narrative. Although other studies may not have used WGS, the genetic diversity and socioeconomic features associated with TB in Ethiopian children have been previously documented. If a goal of this work is to demonstrate the need for broader use of WGS in Ethiopia, then tell that story. Lines 95-96 of the Introduction state that genomic data is "crucial for early detection of resistant strains, guiding treatment decisions, and informing national TB control strategies". The current study shows excellent agreement between pDST and WGS, so what is the need for change? Where is the data that suggests WGS provides improved turnaround times, more comprehensive results and/or is cost effective? If resources are limited, might other WGS approaches (e.g., targeted Nanopore sequencing from sputum) be more suitable for your DST needs? Similarly, if WGS is better than spoligotyping for monitoring lineages, identifying strain clusters and/or ruling out specific transmission events, can you leverage strains or data from previous studies to evaluate that claim? Perhaps some of the strains analyzed in Mollalign et al. 2013. [doi: 10.1371/journal.pone.0284363 "Genetic diversity and drug sensitivity profile of *Mycobacterium tuberculosis* among children in Ethiopia."] were sequenced for the current work?

Additional Comments

Lines 167-169 indicate that DST results were concordant for 23 cases. What about the 3 cases that were only TB-DR by phenotypic testing and the 2 cases that were only TB-DR by WGS analysis? Was pDST repeated? Or was a different method used to determine the 'true' result?

Tables: The values in the tables should be reviewed for accuracy.

Table 1: The % Rural areas is listed as 75.6%, but 60/85 is only 70.6%

Table 5: In the 'Age' category, the '10-14' group includes 47 MDR cases. Also, most categories in Table 5 show a total of 18 MDR cases, whereas the text indicates that 17/85 cases were MDR-TB. Is the 'extra' case a pDST result that was not classified as MDR by WGS?

Reviewer #3 (Comments for the Author):

In this study, the authors used WGS to analyze the Mtb strains isolated from Children in Ethiopia, which gives a good suggestion for integrating genomic surveillance into childhood TB and drug resistance control.

Comments:

1: The title: The species name such as "*Mycobacterium tuberculosis*" needs to be italicized.

2: Abstract: Line 31: "Pre-XDR-TB" not "Per-XDR-TB".

3: The sentence (Lines 38~39) needs to be rephrased.

4: Introduction (Lines 55~61), the authors need to use the new data from the 2025 WHO tuberculosis Report.

5: The ratios of residence in Table 1 need to be checked and revised.

6: In the methods part, the authors may describe more on DST.

Title: Genetic Diversity and Drug Resistance Profile of *Mycobacterium tuberculosis* among Ethiopian Children as Determined by Whole-Genome Sequencing.

Dear Editor,

Thank you very much for allowing us to revise and resubmit our manuscript in response to the reviewers' comments again. We also thank the reviewers for their constructive comments, which have improved the quality and clarity of our work. Below, we provide a detailed point-by-point response to each reviewer's comment. In the revised manuscript, all changes made in response to the reviewers' suggestions are highlighted in track changes for ease of reference.

Reviewer 1:

Comment 1: In the current study, Abebaw and colleagues describe the genetic diversity and drug resistance patterns of *Mycobacterium tuberculosis* among Ethiopian children. This revised manuscript has an updated title and addresses several issues raised by previous reviewers. In their response to the reviewers, the authors also acknowledge that various statistical analyses were compromised by the limited number of samples available for study. Even so, the discussion still mentions "statistically significant associations" between MDR-TB status and specific Mtb lineages. Such trends are more easily explained by sampling biases. There are 12 lineages/sublineages described in the study, but the 2 most common (L-4.2.2.2 and L-3) represent 56% (48/85) of all isolates whereas the 6 least common represent only 10.5% (9/85). Those numbers are too small for a meaningful comparison. For lineages L-2, L-4.1.4 or L-4.4.1, a single resistant isolate would have given scores of 100% MDR.

Response to Comment 1: We agree that the uneven distribution of *Mycobacterium tuberculosis* lineages and the small number of isolates in several sublineages limit lineage-specific inference and may introduce sampling bias. While Lineage 4.2.2.2 was predominant in our dataset, this observation is interpreted in the context of previously reported national TB epidemiology rather than as evidence of lineage-specific drug resistance

We thank the reviewer for this insightful comment. In response, we have revised removed references to statistically significant lineage-specific associations and to clarify that the observed lineage patterns likely reflect background transmission dynamics and country-level TB burden. These findings are now explicitly described in lines 228-234.

Comment 2: Sampling also explains the "high genetic diversity" detected in Oromia and SNNP. Those two regions accounted for >85% of all samples. There is no genetic diversity in Amhara, Dire Dawa or Tigray because each of those regions were only represented by single cases.

Response to Comment 2: We have accepted the reviewer's suggestion and revised the manuscript as "The geographical distribution of the dominant tuberculosis lineages from this study was mainly observed in Oromia (49/85; 57.6%) and the Southern Nations, Nationalities, and Peoples' Region (SNNP) (26/85; 30.5%), as the majority of these cases were collected from these regions." in line 127-129

Comment 3: The authors could consider re-framing their narrative. Although other studies may not have used WGS, the genetic diversity and socioeconomic features associated with TB in Ethiopian children have been previously documented. If a goal of this work is to demonstrate the need for broader use of WGS in Ethiopia, then tell that story. Lines 95-96 of the Introduction state that genomic data is "crucial for early detection of resistant strains, guiding treatment decisions, and informing national TB control strategies". The current study shows excellent agreement between pDST and WGS, so what is the need for change? Where is the data that suggests WGS provides improved turnaround times, more comprehensive results and/or is cost effective? If resources are limited, might other WGS approaches (e.g., targeted Nanopore sequencing from sputum) be more suitable for your DST needs?

Response for comment 3: We have accepted the reviewer's comment and suggestion. We have revised the Introduction (lines 84-96) and the discussion (lines 235-246) to demonstrate the reliability and added value of whole-genome sequencing (WGS) for TB surveillance in Ethiopia. Additionally, we acknowledge the cost-effectiveness and implementation feasibility of WGS as a limitation (line 268-269) and recommend in the Conclusion part to compare it with other alternative targeted sequencing approaches as potential future options (lines 275-280).

Comment 4: Similarly, if WGS is better than spoligotyping for monitoring lineages, identifying strain clusters, and/or ruling out specific transmission events, can you leverage strains or data from previous studies to evaluate that claim? Perhaps some of the strains analyzed in Mollalign et al. 2013. [doi: 10.1371/journal.pone.0284363 "Genetic diversity and drug sensitivity profile of Mycobacterium tuberculosis among children in Ethiopia."] were sequenced for the current work?

Response for comment 4: We agree that whole-genome sequencing provides substantially higher resolution than spoligotyping for monitoring *Mycobacterium tuberculosis* lineages, identifying strain clusters, and evaluating potential transmission events. However, the isolates analyzed in the study by Mollaligh et al. (2023) were characterized using conventional genotyping methods and were not subjected to WGS.

Comment 5: Lines 167-169 indicate that DST results were concordant for 23 cases. What about the 3 cases that were only TB-DR by phenotypic testing and the 2 cases that were only TB-DR by WGS analysis? Was pDST repeated? Or was a different method used to determine the 'true' result?

Response for comment 5: We apologize for the lack of clarity in the original manuscript. No additional molecular testing was conducted for discordant cases; however, phenotypic drug susceptibility testing was repeated for all discordant isolates to confirm the initial results. This has been clarified in the revised Methods section (lines 370–371).

Comment 6: Tables: The values in the tables should be reviewed for accuracy. Table 1: The % Rural areas is listed as 75.6%, but 60/85 is only 70.6%. Table 5: In the 'Age' category, the '10-14' group includes 47 MDR cases. Also, most categories in Table 5 show a total of 18 MDR cases, whereas the text indicates that 17/85 cases were MDR-TB. Is the 'extra' case a pDST result that was not classified as MDR by WGS?

Response to comment 6: We thank the reviewer for carefully reviewing the tables and identifying these inconsistencies. We apologize for the errors and have corrected them in the revised manuscript as below.

Table 1: The percentage of participants from rural areas was incorrectly reported as 75.6%. This has been corrected to 70.6% (60/85). Table 5 (Age category): We confirm that the value “47 MDR cases” in the 10–14 age group was a typographical error. The correct number of MDR cases in this age group is 14, and Table 5 has been revised accordingly. Beside this, one isolate classified as MDR-TB by phenotypic DST was not classified as MDR-TB by WGS. For Table 5, MDR-TB status was defined based on phenotypic DST, which is why 18 cases are shown.

Reviewer 3 comment:

Comment: In this study, the authors used WGS to analyze the *Mtb* strains isolated from Children in Ethiopia, which gives a good suggestion for integrating genomic surveillance into childhood TB and drug resistance control.

Response to Comment: We thank the reviewer for this positive comment and for recognizing the relevance of this work to genomic surveillance of childhood TB in Ethiopia.

Comment 1: The title: The species name such as "*Mycobacterium tuberculosis*" needs to be italicized.

Response to Comment 1:

We apologize for missing to be italicized. The species name *Mycobacterium tuberculosis* has now been italicized in the title.

Comment 2: Abstract: Line 31: "Pre-XDR-TB" not "Per-XDR-TB".

Response to Comment 2: We apologize for this error. The term "Per-XDR-TB" has been corrected to "Pre-XDR-TB" in the Abstract (Line 31).

Comment 3: The sentence (Lines 38~39) needs to be rephrased.

Response to Comment 3: We thank the reviewer for this suggestion. The sentence in Lines 38–41 has been rephrased for clarity and readability in the revised manuscript.

Comment 4: Introduction (Lines 55~61), the authors need to use the new data from the 2025 WHO tuberculosis Report.

Response Comment 4: We have accepted the reviewer's suggestion and used the most recent data from the 2025 WHO Global Tuberculosis Report (lines 58-59).

Comment 5: The ratios of residence in Table 1 need to be checked and revised.

Response to Comment 5: We thank the reviewer for carefully reviewing Table 1 and identifying these inconsistencies. We apologize for the errors and have corrected it to 70.6% (60/85).

Comment 6: Line 136. In the methods part, the authors may describe more on DST.

Response to Comment 6: We have expanded the Methods section to provide additional details on drug susceptibility testing (DST). The revised text now specifies the phenotypic DST methodology, including the drugs tested, concentrations, and interpretation, in accordance with WHO-recommended critical concentrations (lines 331-336).

Re: Spectrum02736-25R2 (**Genetic Diversity and Drug Resistance Profiles of Mycobacterium tuberculosis among Ethiopian Children as Determined by Whole-Genome Sequencing**)

Dear Mrs. Yeshiwork Abebaw:

Your manuscript has been accepted, and I am forwarding it to the ASM production staff for publication. Your paper will first be checked to make sure all elements meet the technical requirements. ASM staff will contact you if anything needs to be revised before copyediting and production can begin. Otherwise, you will be notified when your proofs are ready to be viewed.

Sincerely,
Florence Doucet-Populaire
Editor
Microbiology Spectrum